# EPIC: Compressing Deep GNNs via Expressive Power Gap-Induced Knowledge Distillation

## Abstract

The teacher-student paradigm-based knowledge distillation (KD) has recently emerged as a promising technique for compressing graph neural networks (GNNs). Despite the great success in compressing moderate-sized GNNs, distilling deep GNNs (e.g., with over 100 layers) remains a tough challenge. A widely recognized reason is the *teacher-student expressive power gap*, i.e., the embeddings of a deep teacher may be extremely hard for a shallow student to approximate. Besides, the theoretical analysis and measurement of this gap are currently missing, resulting in a difficult trade-off between the needs of being "lightweight" and being "expressive" when selecting a student for the deep teacher. To bridge the theoretical gap and address the challenge of distilling deep GNNs, we propose the *first* GNN KD framework that quantitatively analyzes the teacher-student expressive power gap, namely **E**xpressive **P**ower gap-**I**ndu**C**ed knowledge distillation (**EPIC**). Our key idea is to formulate the estimation of the expressive power gap as an embedding regression problem based on the theory of polynomial approximation. Then, we show that the minimum approximation error has an upper bound, which decreases rapidly with respect to the number of student layers. Furthermore, we empirically demonstrate that the upper bound exponentially converges to zero as the number of student layers increases. Moreover, we propose to narrow the range of tuning the number of student layers based on the upper bound, and use an expressive power gap-related loss term to further encourage the student to generate embeddings similar to those of the teacher. Experiments on large-scale benchmarks demonstrate that EPIC can effectively reduce the numbers of layers of deep GNNs, while achieving comparable or superior performance. Specifically, for the 1,001-layer RevGNN-Deep, we reduce the number of layers by 94% and accelerate inference by roughly eight times, while achieving comparable performance in terms of ROC-AUC on the large-scale benchmark `ogbn-proteins`.

## 1 Introduction

Graph neural networks (GNNs) have recently achieved great success in many real-world applications involving graph data, such as electronic design automation (Ghose et al., 2021; Wang et al., 2022b), combinatorial optimization (Cappart et al., 2023; Peng et al., 2021), search engines (Zhu et al., 2021), recommendation systems (Fan et al., 2019; Wu et al., 2022c), and molecular property prediction (Wieder et al., 2020; Wang et al., 2022a). The key idea of GNNs is to iteratively update node embeddings based on both node features and graph structures (Gilmer et al., 2017; Hamilton, 2020; Kipf & Welling, 2017). Specifically, at each layer, GNNs aggregate messages from each node's neighborhood and then update node embeddings based on aggregation results and node features. Thus, the final embedding of a node in an $L$-layer GNN contains the information about its $L$-hop neighborhood (Hamilton, 2020).

Recently, deep GNNs (Li et al., 2021; Liu et al., 2020; Chen et al., 2020; Li et al., 2019; 2020) have shown leading performance on large-scale public benchmarks such as Microsoft Academic Graph (MAG) (Wang et al., 2020) and Open Graph Benchmark (OGB) (Hu et al., 2020), due to their powerful ability to learn long-range interactions (Chen et al., 2022; Cong et al., 2021). Many techniques improve the prediction performance of deep GNNs from aspects of graph convolutions (Li et al., 2021; Chen et al., 2020), normalization (Zhao & Akoglu, 2019; Guo et al., 2023a; Zhou et al., 2020; 2021), and initialization (JAISWAL et al., 2022) to address the challenges of over-

smoothing (Li et al., 2019; Zeng et al., 2021), over-squashing (Topping et al., 2022), and information bottleneck (Alon & Yahav, 2021).

Nevertheless, the large numbers of layers in deep GNNs severely slow down their inference, making it difficult to deploy deep GNNs in latency-limited scenarios (Chen et al., 2021; Lee et al., 2019; Gong et al., 2021). Unlike models without graph dependency (e.g., Multi-layer perceptrons/Transformers (Vaswani et al., 2017)/CNNs (Krizhevsky et al., 2012)), deep GNNs suffer from the notorious neighbor explosion issue incurred by data dependency (Hamilton et al., 2017), i.e., the exponentially increasing dependencies of nodes with the number of layers. Therefore, knowledge distillation—a promising class of techniques for accelerating inference, while maintaining prediction performance—have recently become popular in real applications of GNNs.

Various knowledge distillation techniques for GNNs (KD4GNN) have achieved great success in compressing moderate-sized or shallow GNNs (e.g., GCN (Kipf & Welling, 2017), GraphSAGE (Hamilton et al., 2017), GAT (Veličković et al., 2018), and APPNP (Gasteiger et al., 2018). Knowledge distillation aims to train a lightweight "student" model under the supervision of a well-trained "teacher" model, where the supervisory signal (i.e., knowledge) can be anything computed by the teacher model (Gou et al., 2021). Most existing KD4GNN methods (Yang et al., 2022; Guo et al., 2023b; Yang et al., 2020b; Wu et al., 2022a; Zhang et al., 2020b; He et al., 2022) aim to distill GNNs into GNNs with fewer layers or parameters by improving the training framework, the selection of knowledge, or the configurations of teacher models. Another line of impressive works (Yang et al., 2021; Zhang et al., 2021; Tian et al., 2022) propose to distill GNNs into multi-layer perceptrons (MLPs) to avoid the dependency on graph structural information, e.g., neighbor nodes and edge features. The aforementioned works take the significant first step in exploring KD4GNN, and thus have drawn increasing attention in recent years.

However, distilling deep GNNs (e.g., with over 100 layers) remains a tough challenge. A widely recognized reason is that knowledge distillation may be ineffective when the *expressive power gap* between the teacher and student is large, as the embeddings of the teacher may be extremely hard for the student to approximate (Mirzadeh et al., 2020; Gao et al., 2021; Li & Leskovec, 2022). A typical example is distilling GNNs to MLPs. Because MLPs cannot distinguish two nodes with the same features and different neighbors, their expressive power is much weaker than GNNs, hence they are not "good students" for distilling deep GNNs. Besides, the theoretical analysis and measurement of the expressive power gap are currently missing, so it is difficult for us to select for the deep teacher a student that is shallow, while expressive.

In this paper, we aim to bridge the theoretical gap and address the challenge of distilling deep GNNs that *excel on large-scale graphs* and *possess numerous layers*. Towards this goal, we propose the *first* KD4GNN framework that quantitatively analyzes the teacher-student expressive power gap, namely **E**xpressive **P**ower gap-**I**ndu**C**ed Knowledge Distillation (**EPIC**). Our key idea is to formulate the estimation of the expressive power gap as an embedding regression problem based on the theory of polynomial approximation. Then, we show that the minimum approximation error has an upper bound, i.e., the EPIC bound, which decreases rapidly with respect to the number of student layers. Furthermore, our numerical experiments demonstrate that the EPIC bound exponentially converges to zero as the number of student layers increases, which empirically guarantees that we can distill deep GNNs into shallow GNNs. Moreover, we propose to narrow the range of tuning the number of student layers based on the EPIC bound, and use an expressive power gap-related loss to further encourage the student to generate embeddings similar to those of the teacher. Experiments on large-scale benchmarks demonstrate that EPIC can effectively reduce the numbers of layers of deep GNNs, while achieving comparable or superior performance. Specifically, for the 1,001-layer RevGNN-Deep, we reduce the number of layers by 94% and accelerate inference by roughly eight times, while achieving comparable performance in terms of ROC-AUC on the large-scale benchmark `ogbn-proteins`.

## 2 RELATED WORK

In this section, we discuss some works related to our proposed framework.

### 2.1 DEEP GRAPH NEURAL NETWORKS

Deep GNNs have strong potential to complete tasks involving large-scale graph-structured data due to their powerful ability to learn long-range interactions (Chen et al., 2022; Cong et al., 2021). Many works propose various approaches, including skip connection (Li et al., 2019; 2020; Xu et al., 2018;

Gasteiger et al., 2018; Chen et al., 2020), graph normalization (Ioffe & Szegedy, 2015; Zhao & Akoglu, 2019; Zhou et al., 2021; Yang et al., 2020a; Zhou et al., 2020), random dropping (Rong et al., 2019; Huang et al., 2020), and grouped reversible graph connections (Li et al., 2021), to train powerful deep GNNs on large-scale graphs (Chen et al., 2022). Various deep GNNs (Li et al., 2021; Liu et al., 2020; Chen et al., 2020; Li et al., 2019; 2020) have shown leading performance on large-scale public benchmarks such as Microsoft Academic Graph (MAG) (Wang et al., 2020) and Open Graph Benchmark (OGB) (Hu et al., 2020). It is worth mentioning that the 1,001-layer RevGNN-Deep trained on `ogbn-proteins` is the deepest GNN that has been published so far. Despite the great success on public benchmarks, deep GNNs suffer from the notorious neighbor explosion problem incurred by the data dependency (Hamilton et al., 2017), i.e., the exponentially increasing dependencies of nodes with the number of layers. This poses significant challenges to employing deep GNNs in latency-limited scenarios. Therefore, we aim to compress deep GNNs via the promising teacher-student paradigm-based knowledge distillation (KD) techniques.

## 2.2 KNOWLEDGE DISTILLATION FOR GNNS (KD4GNN)

To compress GNNs, many works propose various KD (Hinton et al., 2015) techniques, which aims to train lightweight "student" models under the supervision of well-trained "teacher" models.

**GNNs-to-GNNs Distillation.** Most existing KD4GNN works aim to distill GNNs to GNNs with fewer layers or parameters. According to their key ideas for improving the distillation, we can classify them into three categories: (1) what to distill (how to select appropriate knowledge) (Yang et al., 2022; Huo et al., 2023; Yan et al., 2020; Yang et al., 2020b; Zhang et al., 2020b; He et al., 2022), (2) how to distill (how to improve the training paradigm to better transfer knowledge) (Guo et al., 2023b; Zhang et al., 2020b; He et al., 2022), and (3) who will teach (how to select appropriate teachers) (Huo et al., 2023; Guo et al., 2023b; Zhang et al., 2020b).

**GNNs-to-MLPs Distillation.** Another line of impressive works (Yang et al., 2021; Zhang et al., 2021; Tian et al., 2022; Wu et al., 2023) propose to distill GNNs into multi-layer perceptrons (MLPs) to avoid the dependency on graph structural information, such as neighbor nodes and edge features. Because MLPs take only node features as input, their inference speed is much faster than GNNs. However, as shown in Section 6.2, the distillation performance of these methods in the inductive setting is unsatisfactory, as the expressive power of MLPs is much weaker than GNNs and MLPs do not have access to the teachers' soft labels on test nodes to use as guidance.

**Distillation for Deep GNNs.** The aforementioned works have achieved great success in distilling moderate-sized or shallow GNNs (e.g., GCN (Kipf & Welling, 2017), GraphSAGE (Hamilton et al., 2017), GAT (Veličković et al., 2018), and APPNP (Gasteiger et al., 2018). However, distilling deep GNNs (e.g., with over 100 layers) remains a tough challenge. A widely recognized reason is that KD may be ineffective when the expressive power gap between the teacher and student is large, as the embeddings of the teacher may be extremely hard for the student to approximate (Mirzadeh et al., 2020; Gao et al., 2021; Li & Leskovec, 2022). Besides, the theoretical analysis and measurement of this gap are currently missing. Thus, we aim to bridge the theoretical gap and address the challenge of distilling deep GNNs that *excel on large-scale graphs* and *possess numerous layers*. To the best of our knowledge, this is the *first* paper that attempts to distill GNNs with more than 100 layers.

## 2.3 EXPRESSIVE POWER OF GRAPH NEURAL NETWORKS

The expressive power of GNNs has attracted much attention recently. For example, (Xu et al., 2019) and (Morris et al., 2019) show that message passing-based GNNs are at most as powerful as 1-WL test (Weisfeiler & Leman, 1968) to distinguish non-isomorphic graphs. Thus, many works propose various techniques to increase the power of GNNs from the perspective of higher-order WL tests (Morris et al., 2019; Maron et al., 2019; Chen et al., 2019) and graph bi-connectivity (Zhang et al., 2023). However, the above-mentioned works mainly focus on GNNs' whole-graph expressive power. Therefore, to analyze the GNNs' link expressive power, (Zhang et al., 2020a) proposes a multi-node representation theory and explains the superior performance of the labeling trick used in GNNs for link prediction. Besides, for node property prediction, (Wang & Zhang, 2022) shows that spectral GNNs can produce arbitrary graph signals under some mild conditions.

## 3 PRELIMINARIES

We introduce notations, GNNs, and KD in Sections 3.1, 3.2, and 3.3, respectively.

### 3.1 NOTATIONS

A graph $\mathcal{G} = (\mathcal{V}, \mathcal{E})$ is defined by an unordered set of nodes $\mathcal{V} = \{v_1, v_2, \ldots, v_n\}$, where $n$ is the number of nodes, and a set of edges $\mathcal{E} \subset \mathcal{V} \times \mathcal{V}$ among these nodes. The node set $\mathcal{V} = \mathcal{V}^{\mathrm{L}} \sqcup \mathcal{V}^{\mathrm{U}}$ is the disjoint union of the labeled node set $\mathcal{V}^{\mathrm{L}}$ and the unlabeled node set $\mathcal{V}^{\mathrm{U}}$. The label of a node $v_i \in \mathcal{V}^{\mathrm{L}}$ is $y_i$. Let $(v_i, v_j) \in \mathcal{E}$ denote an edge going from $v_i \in \mathcal{V}$ to $v_j \in \mathcal{V}$ and $\mathcal{N}(v_i) = \{v_j \in \mathcal{V} | (v_i, v_j) \in \mathcal{E}\}$ denote the neighborhood of $v_i$. Let $\mathbf{A} \in \{0, 1\}^{n \times n}$ be the adjacency matrix of $\mathcal{G}$ ($\mathbf{A}_{i,j} = 1$ if and only if $(v_i, v_j) \in \mathcal{E}$, $i, j \in [n]$) and $\mathbf{D}$ be the diagonal matrix whose diagonal element $\mathbf{D}_{i,i}$ is the degree of $v_i$, $i \in [n]$. We assume that $\mathcal{G}$ is undirected, i.e., $v_j \in \mathcal{N}(v_i) \Leftrightarrow v_i \in \mathcal{N}(v_j)$, hence $\mathbf{A}$ is symmetric. Denote the normalized adjacency matrix by $\widehat{\mathbf{A}} = \mathbf{D}^{-\frac{1}{2}} \mathbf{A} \mathbf{D}^{-\frac{1}{2}}$. Let $\mathbf{I}$ be the identity matrix, and denote the normalized Laplacian matrix by $\widehat{\mathbf{L}} = \mathbf{I} - \widehat{\mathbf{A}}$, whose eigendecomposition is $\widehat{\mathbf{L}} = \mathbf{U} \mathbf{\Lambda} \mathbf{U}^{\top}$, where $\mathbf{U}$ is the orthogonal matrix of eigenvectors and $\mathbf{\Lambda}$ is the diagonal matrix of eigenvalues. In some scenarios, nodes are associated with a node feature matrix $\mathbf{X} \in \mathbb{R}^{n \times d_x}$. For a positive integer $L$, $[L]$ denotes $\{1, \ldots, L\}$. For $i \in [n]$, let $\mathbf{X}_{i,:} \in \mathbb{R}^{d_x}$ and $\mathbf{H}_{i,:} \in \mathbb{R}^{d_h}$ denote the feature and the embedding of the node $v_i$ with dimension $d_x$ and $d_h$, respectively. Let $\mathbf{H} = \left(\mathbf{H}_{1,:}^{\top}, \ldots, \mathbf{H}_{n,:}^{\top}\right)^{\top} \in \mathbb{R}^{n \times d_h}$ be the embedding matrix of the graph. We also denote the embeddings of a set of nodes $\mathcal{S} = \{v_{i_k}\}_{k=1}^{|\mathcal{S}|}$ by $\mathbf{H}_{\mathcal{S}} = (\mathbf{H})_{\mathcal{S}} = \left(\mathbf{H}_{i_1,:}^{\top}, \ldots, \mathbf{H}_{i_k,:}^{\top}\right)^{\top} \in \mathbb{R}^{d_h \times |\mathcal{S}|}$. For a matrix $\mathbf{W} \in \mathbb{R}^{p \times q}$, we denote its $i$-th row by $\mathbf{W}_{i,:}$ and $j$-th column by $\mathbf{W}_{:,j}$, where $i \in [p]$ and $j \in [q]$, respectively.

### 3.2 GRAPH NEURAL NETWORKS

In this paper, we focus on node-level tasks on graphs, which aim to predict a discrete or continuous label for each node. Graph neural networks (GNNs) iteratively update node embeddings based on node features and graph structures (Hamilton, 2020; Kipf & Welling, 2017). At each layer, GNNs aggregate messages from each node's neighborhood and then update node embeddings based on aggregation results and node features.

A GNN with $L$ layers and parameters $(\theta^{(l)})_{l=1}^{L}$ generates the final node embeddings $\mathbf{H} = \mathbf{H}^{(L)}$ as

$$\mathbf{H}^{(l)} = f_{\theta^{(l)}}(\mathbf{H}^{(l-1)}; \mathbf{X}, \mathbf{A}), \ l \in [L], \tag{1}$$

where $\mathbf{H}^{(0)} = \mathbf{X}$ and $f_{\theta^{(l)}}$ is the $l$-th layer with parameters $\theta^{(l)}$.

### 3.3 KNOWLEDGE DISTILLATION

Knowledge Distillation (KD) aims to train a lightweight student model $S$ by transferring the knowledge from a well-trained teacher model $T$ (Tian et al., 2023). The key idea of KD is to encourage $S$ to mimic the behaviors of $T$ under its supervision, where the supervisory signal (i.e., knowledge) can be anything computed by $T$, such as logits and hidden embeddings (Gou et al., 2021).

Specifically, given the knowledge $k_T$ and $k_S$ computed by $T$ and $S$, respectively, we define the knowledge distillation loss as $\mathcal{L}_{kd} = \mathrm{dist}(k^T, k^S)$, where $\mathrm{dist}(\cdot, \cdot)$ is a distance function (note that we do not require it to be symmetric), such as the Kullback-Leibler divergence and the Euclidean distance. For example, for the vanilla knowledge distillation proposed in (Hinton et al., 2015), the knowledge is the soft labels derived by applying the Softmax operation on logits, and the distance function is the Kullback-Leibler divergence.

## 4 THEORETICAL ANALYSIS OF EXPRESSIVE POWER GAP

In the knowledge distillation of deep GNNs, a common but knotty problem is how to select an appropriate value for the number of student layers, given a deep teacher. This problem involves another one that is more essential, i.e., *how does the number of layers of one GNN affect its expressive power?* Specifically, deep $L$-layer teacher GNNs encode the information about nodes' $L$-hop neighbors (Hamilton, 2020), while shallow $M$-layer student GNNs are difficult to encode the long-range interactions as well as the deep teacher if $M \ll L$. To address this problem, we theoretically analyze the gap between the expressive power of GNNs with different numbers of layers in this section.

We analyze the expressive power of GNNs on node-level tasks from a spectral perspective following (Wang & Zhang, 2022). Given a graph $\mathcal{G}$ with node features $\mathbf{X} \in \mathbb{R}^{n \times d_x}$ and the normalized Laplacian matrix $\widehat{\mathbf{L}} = \mathbf{U} \mathbf{\Lambda} \mathbf{U}^{\top}$, where $\mathbf{\Lambda} = \mathrm{diag}(\lambda_1, \ldots, \lambda_n)$ and $\mathbf{U} \in \mathbb{R}^{n \times n}$ is an orthogonal matrix, a spectral-based $L$-layer GNN generates node embeddings at the $l$-th layer as

$$\mathbf{H}^{(l)} = g_{\mathbf{a}}^{(l)}(\widehat{\mathbf{L}}) \phi_{\mathbf{w}}(\mathbf{X}), \ l \in [L], \tag{2}$$

where $g_{\mathbf{a}}^{(l)}(t) = \sum_{k=0}^{l} a_k t^k$ is an $l$-degree polynomial with parameters $\mathbf{a} = (a_k)_{k=0}^{l}$ and $\phi_{\mathbf{w}}$ is a neural network with parameters $\mathbf{w}$. In the theoretical analysis, we suppose that the following assumptions

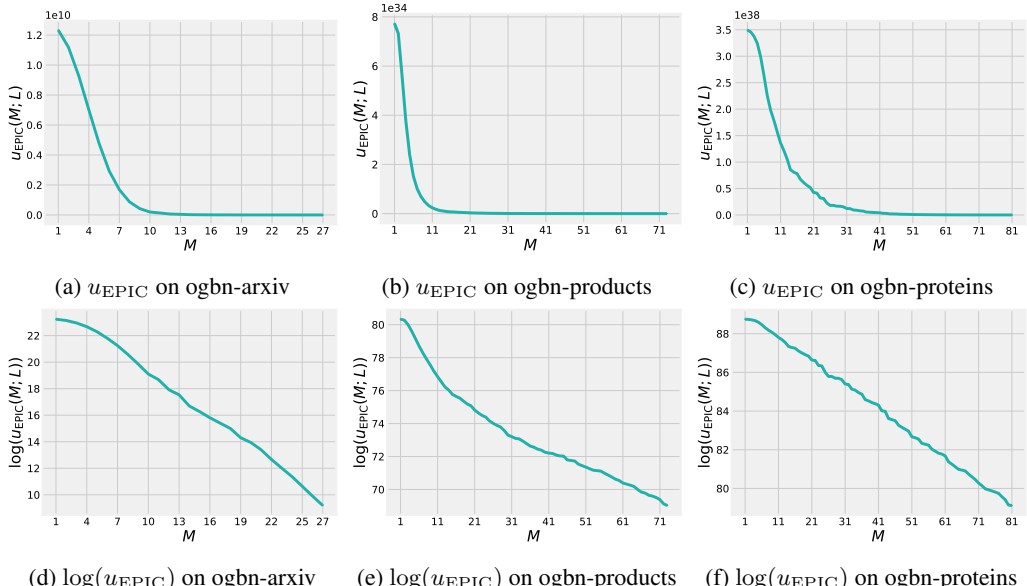

(a) $u_{\mathrm{EPIC}}$ on ogbn-arxiv  (b) $u_{\mathrm{EPIC}}$ on ogbn-products  (c) $u_{\mathrm{EPIC}}$ on ogbn-proteins

(d) $\log(u_{\mathrm{EPIC}})$ on ogbn-arxiv  (e) $\log(u_{\mathrm{EPIC}})$ on ogbn-products  (f) $\log(u_{\mathrm{EPIC}})$ on ogbn-proteins

Figure 1: $u_{\mathrm{EPIC}}(M; L)$ and $\log(u_{\mathrm{EPIC}}(M; L))$ with respect to $M$ on three large-scale datasets.

hold in this paper. For the relationship of our spectral-based theoretical analysis to non-linear GNNs, please refer to Appendix B.1.

**Assumption 1.** *We assume that (1) the $n$ eigenvalues $(\lambda_i)_{i=1}^n$ of $\widehat{\mathbf{L}}$ are different from each other, (2) there exists $C > 0$ such that $\|\phi_{\mathbf{w}}(\mathbf{X})\|_F < C/n^2$ for any $\mathbf{w}$.*

Given a well-trained $L$-layer GNN denoted $G_T^{(L)}$ with embeddings $\mathbf{H}_T^{(L)} \in \mathbb{R}^{n \times d_h}$ and an $M$-layer GNN denoted $G_S^{(M)}$ with embeddings $\mathbf{H}_S^{(M)} \in \mathbb{R}^{n \times d_h}$, where $M < L$, we next analyze their expressive power gap. Suppose that the embeddings are computed by

$$\mathbf{H}_T^{(L)} = g_{\mathbf{a}}^{(L)} \phi_{\mathbf{w}_T}(\mathbf{X}), \quad \mathbf{H}_S^{(M)} = g_{\mathbf{b}}^{(M)} \phi_{\mathbf{w}_S}(\mathbf{X}), \tag{3}$$

respectively. We formulate the estimation of their expressive power gap as finding the minimum approximation error of $\mathbf{H}_S^{(M)}$ to $\mathbf{H}_T^{(L)}$ in terms of the Frobenius norm, i.e.,

$$e(M; L) \triangleq \min_{\mathbf{b}, \mathbf{w}_S} \|\mathbf{H}_S^{(M)} - \mathbf{H}_T^{(L)}\|_F, \tag{4}$$

The following theorem gives an upper bound of $e(M; L)$ and shows that the upper bound decreases monotonically with respect to $M$. For the detailed proof, please refer to Appendix B.

**Theorem 1.** *Given an $n$-node graph $\mathcal{G}$ with node features $\mathbf{X}$ and the normalized Laplacian matrix $\widehat{\mathbf{L}} = \mathbf{U}\mathbf{\Lambda}\mathbf{U}^\top$, an $L$-layer well-trained GNN ($G_T^{(L)}$) and an $M$-layer GNN ($G_S^{(M)}$) that compute embeddings as shown in Eq. (3), we suppose that Assumption 1 holds, then we have*

$$e(M; L) \leq u_{\mathrm{EPIC}}(M; L) \triangleq C\|\mathbf{P}^{(M)}(\mathbf{P}^{(M)})^\top \mathbf{d}^{(M)} - \mathbf{d}^{(M)}\|_2, \tag{5}$$

*where $e(M; L)$ is defined by Eq. (4), $u_{\mathrm{EPIC}}(M; L)$ is named the **EPIC bound**, and $\mathbf{P}^{(M)} \in \mathbb{R}^{n \times (M+1)}$ is the left-singular matrix of*

$$\mathbf{V}^{(M)} = \begin{pmatrix} 1 & \lambda_1 & \cdots & \lambda_1^M \\ \vdots & \vdots & \ddots & \vdots \\ 1 & \lambda_n & \cdots & \lambda_n^M \end{pmatrix}, \tag{6}$$

*and*

$$\mathbf{d}^{(M)} = \left( \sum_{k=M+1}^L a_k \lambda_1^k, \ldots, \sum_{k=M+1}^L a_k \lambda_n^k \right)^\top. \tag{7}$$

*Moreover, $u_{\mathrm{EPIC}}(M; L)$ decreases monotonically with respect to $M$. Specifically, we have $u_{\mathrm{EPIC}}(L; L) = 0$, which leads to $e(L; L) = 0$.*

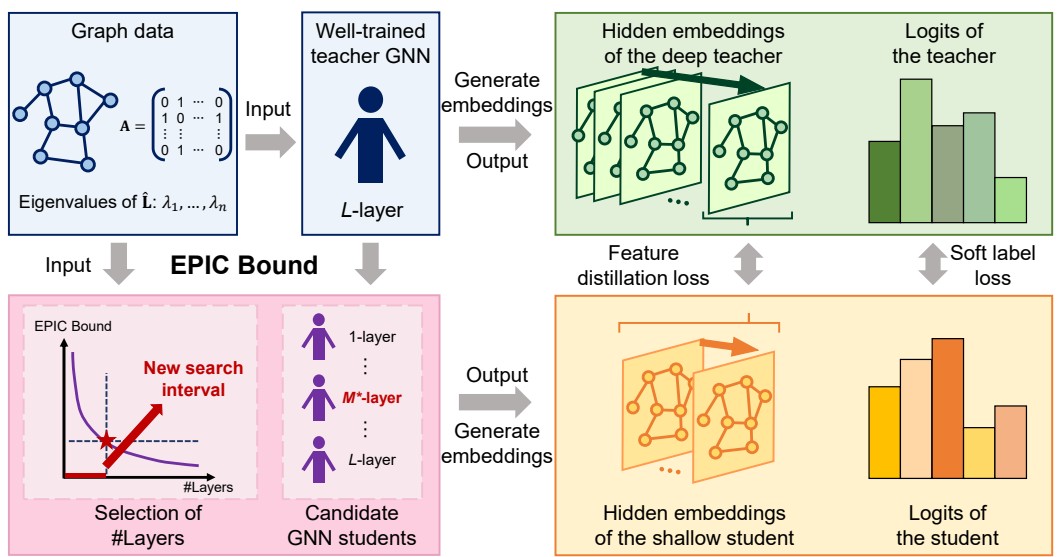

Figure 2: The overall framework of EPIC. Given a graph $\mathcal{G}$ and a well-trained $L$-layer teacher GNN, we compute EPIC bounds for different values of numbers of student layers and narrow the range of tuning the number of student layers (see Section 5.1). Then, we use an expressive power gap-related loss (i.e., a feature distillation loss) to further encourage the student to generate embeddings similar to those of the teacher (see Section 5.2).

Please note that the monotonically decreasing property of $u_{\text{EPIC}}(M; L)$ does not depend on $L$ and $\mathbf{a} = (a_k)_{k=0}^L$. To further study how the EPIC bound $u_{\text{EPIC}}(M; L)$ decreases as $M$ increases, we conduct numerical experiments on large-scale datasets `ogbn-arxiv`, `ogbn-products`, and `ogbn-proteins`. As shown in Figure 1, $u_{\text{EPIC}}(M; L)$ converges exponentially to zero (note that $u_{\text{EPIC}}(L; L) = 0$) as $M$ increases. This empirically guarantees that we can distill deep GNNs into shallow GNNs. For more details about the experiments, please refer to Section 6.4.

## 5 EXPRESSIVE POWER GAP-INDUCED KNOWLEDGE DISTILLATION

Based on our theoretical analysis in Section 4, we propose a GNN KD framework that takes the teacher-student expressive power gap into account, namely Expressive Power Gap-Induced Knowledge Distillation (EPIC). Figure 2 illustrates the overall framework of EPIC. Specifically, given a graph $\mathcal{G}$ and a well-trained deep GNN with $L$ layers, we first compute EPIC bounds for different values of numbers of student layers and select an appropriate value for the number of student layers such that the student is shallow, while expressive. Then, to further encourage the student to generate embeddings similar to those of the teacher, we propose an expressive power gap-induced loss term.

### 5.1 SELECTION OF NUMBER OF STUDENT LAYERS

**Pre-processing: Computing EPIC Bounds.** Given an $L$-layer teacher GNN ($G_T^{(L)}$) and an $n$-node training graph $\mathcal{G}$ with the adjacency matrix $\mathbf{A} \in \mathbb{R}^{n \times n}$, we first compute the normalized Laplacian $\widehat{\mathbf{L}} = \mathbf{I} - \mathbf{D}^{-\frac{1}{2}} \mathbf{A} \mathbf{D}^{-\frac{1}{2}} \in \mathbb{R}^{n \times n}$, where $\mathbf{D}$ is the degree matrix, and the eigenvalues $(\lambda_i)_{i=1}^n$ of $\widehat{\mathbf{L}}$.

Then, for each $M \in [L-1]$, we compute the matrix $\mathbf{V}^{(M)} \in \mathbb{R}^{n \times (M+1)}$ defined by Eq. (6) and its singular value decomposition (SVD) $\mathbf{V}^{(M)} = \mathbf{P}^{(M)} \mathbf{\Sigma}^{(M)} (\mathbf{Q}^{(M)})^\top$, where $\mathbf{P}^{(M)} \in \mathbb{R}^{n \times (M+1)}$, $\mathbf{\Sigma}^{(M)} \in \mathbb{R}^{(M+1) \times (M+1)}$, and $\mathbf{Q}^{(M)} \in \mathbb{R}^{(M+1) \times n}$. If $G_T^{(L)}$ is a spectral GNN, we use its real parameters as $\mathbf{a} = (a_k)_{k=0}^L$. Otherwise, we let $a_k = 1$ for $k = 0, \ldots, L$. Note that this does not affect the monotonically decreasing property of the EPIC bound. Then, we compute the vector $\mathbf{d}$ defined by Eq. (7) and the EPIC bound $u_{\text{EPIC}}(M; L)$ by Eq. 5 (we can simply let $C = 1$).

It is worth noting that we do not need to compute EPIC bounds in the inference stage of GNNs, hence the computation of EPIC bounds do not affect the inference speed of GNNs, which is the focus of this paper and most knowledge distillation methods.

**Narrowing the Range of Tuning the Number of Student Layers.** After computing EPIC bounds $(u_{\text{EPIC}}(M; L))_{M=1}^{L-1}$, we plot $u_{\text{EPIC}}(M; L)$ as a function of $M$. By observing the plot, we find the

maximum value $M_{\max}$ such that $u_{\mathrm{EPIC}}(M;L)$ decreases slowly when $M > M_{\max}$. Then we tune the number $M$ of student layers in $[1, M_{\max}]$, which is much smaller than $[1, L]$ since $u_{\mathrm{EPIC}}(M;L)$ converges exponentially to zero as $M$ increases.

## 5.2 EXPRESSIVE POWER GAP-RELATED LOSS

To further improve the expressive power of the student $G_S^{(M)}$, we concatenate its embeddings at top-$K$ layers and multiply it by a linear mapping matrix $\mathbf{W} \in \mathbb{R}^{Kd_S \times d_T}$ as

$$\widehat{\mathbf{H}}_S^{(M)} = \begin{bmatrix} \mathbf{H}_S^{(M-K+1)} & \cdots & \mathbf{H}_S^{(M)} \end{bmatrix} \mathbf{W} \in \mathbb{R}^{n \times Kd_T}, \tag{8}$$

where $K$ is a hyperparameter, $d_S$ is the width of $G_S^{(M)}$, and $d_T$ is the width of the teacher $G_T^{(L)}$. We take $\widehat{\mathbf{H}}_S^{(M)}$ as the final embedding matrix of $G_S^{(M)}$.

To encourage $G_S^{(M^*)}$ to generate embeddings similar to those of $G_T^{(L)}$, we use an expressive power gap-related loss (feature distillation loss), i.e.,

$$\mathcal{L}_{\mathrm{EP}} = \|(\widehat{\mathbf{H}}_S^{(M)})_{\mathcal{V}^{\mathrm{tr}}} - (\mathbf{H}_T^{(L)})_{\mathcal{V}^{\mathrm{tr}}}\|_F^2, \tag{9}$$

where $\mathcal{V}^{\mathrm{tr}}$ is the train set. Besides, we also use the ground truth loss and the soft label loss, i.e.,

$$\mathcal{L}_{\mathrm{GT}} = \sum_{v \in \mathcal{V}^{\mathrm{tr}}} \mathrm{CE}(\widehat{\mathbf{y}}_v, \mathbf{y}_v), \quad \mathcal{L}_{\mathrm{SL}} = \sum_{v \in \mathcal{V}^{\mathrm{tr}}} D_{\mathrm{KL}}(\widehat{\mathbf{y}}_v, \mathbf{z}_v), \tag{10}$$

where $\mathrm{CE}(\cdot, \cdot)$ is the cross-entropy function, $D_{\mathrm{KL}}(\cdot, \cdot)$ is the Kullback-Leibler divergence, $\widehat{\mathbf{y}}$ is the prediction of $G_S^{(M)}$, $\mathbf{y}$ is the vector of ground truth labels, and $\mathbf{z}$ is the vector of soft labels predicted by $G_T^{(L)}$. The final loss function $\mathcal{L}$ is the weighted sum of the three loss terms, i.e.,

$$\mathcal{L} = \mathcal{L}_{\mathrm{GT}} + \lambda \mathcal{L}_{\mathrm{SL}} + \mu \mathcal{L}_{\mathrm{EP}}, \tag{11}$$

where $\lambda$ and $\mu$ are the weight coefficients of loss terms. We summarize EPIC in Algorithm 1. Please note that it does not involve the process of updating parameters.

## 6 EXPERIMENTS

We first introduce experimental settings in Section 6.1. We then show the main results of EPIC on distilling deep GNNs on large-scale benchmarks in Section 6.2. After that, in Section 6.3, we conduct experiments to study the decreasing trend of the EPIC bound. Finally, we provide ablation studies in Section 6.4. We run all experiments on a single GeForce RTX 3090 Ti (24 GB).

### 6.1 EXPERIMENTAL SETTINGS

We conduct all experiments in the practical setting of the *inductive* training, i.e., test nodes are strictly unseen during training (Wu et al., 2022b). Specifically, given a graph $\mathcal{G} = (\mathcal{V}, \mathcal{E})$ with node features $\mathbf{X}$ and ground truth labels $\mathbf{y}$, where $\mathcal{V} = \mathcal{V}^{\mathrm{L}} \sqcup \mathcal{V}^{\mathrm{U}}$ is the disjoint union of the labeled node set $\mathcal{V}^{\mathrm{L}}$ and the unlabeled node set $\mathcal{V}^{\mathrm{U}}$, we remove all edges connected to nodes in $\mathcal{V}^{\mathrm{U}}$ from $\mathcal{E}$ to

---

**Algorithm 1** EPIC: Expressive Power Gap-Induced Knowledge Distillation

---

1: **Input:** The training graph $\mathcal{G}$ with nodes $\mathcal{V}^{\mathrm{tr}}$, node features $\mathbf{X}$, edge features $\mathbf{E}$ (if $\mathcal{G}$ has them), and the adjacency matrix $\mathbf{A}$. The ground truth labels $(\mathbf{y})_{\mathcal{V}^{\mathrm{tr}}}$, soft labels $(\mathbf{z})_{\mathcal{V}^{\mathrm{tr}}}$ and embeddings $(\mathbf{H}_T^{(L)})_{\mathcal{V}^{\mathrm{tr}}}$ of the teacher $G_T^{(L)}$. The well-trained parameters $\mathbf{a} = (a_k)_{k=0}^L$ (if $G_T^{(L)}$ is spectral). The weight coefficients $\lambda$ and $\mu$. The number $K$ of concatenated embeddings and ratio $\gamma$.
2: Compute $\widehat{\mathbf{L}} = \mathbf{I} - \mathbf{D}^{-\frac{1}{2}} \mathbf{A} \mathbf{D}^{-\frac{1}{2}}$ and eigenvalues $(\lambda_i)_{i=1}^n$
3: **for** $M = 1, \ldots, L-1$ **do**
4:     Compute $\mathbf{V}^{(M)}$ by Eq. (6)
5:     Compute the SVD by $\mathbf{V}^{(M)} = \mathbf{P}^{(M)} \mathbf{\Sigma}^{(M)} (\mathbf{Q}^{(M)})^\top$
6:     Compute $\mathbf{d}^{(M)}$ by Eq. (7)
7:     Compute $u_{\mathrm{EPIC}}(M;L)$ by Eq. (5)
8: **end for**
9: Find $M_{\max}$ by observing the plot of $u_{\mathrm{EPIC}}(M;L)$
10: **for** $M = 1, \ldots, M_{\max}$ **do**
11:     Compute $\widehat{\mathbf{H}}_S^{(M)}$ by Eq. (8)
12:     Compute $\mathcal{L}_{\mathrm{EP}}, \mathcal{L}_{\mathrm{GT}}, \mathcal{L}_{\mathrm{SL}}$, and $\mathcal{L}$ by Eqs. (9)-(11)
13: **end for**

---

form $\mathcal{E}^{\mathrm{L}}$ and $\mathcal{G}^{\mathrm{L}} = (\mathcal{V}^{\mathrm{L}}, \mathcal{E}^{\mathrm{L}})$. For details of the inductive setting, please refer to Appendix A.2.

For the *transductive* setting, instead of training a student model, we can store the final node embedding matrix $\mathbf{H}_T^{(L)}$ of teachers in cheap RAM or hard drive storage to accelerate inference, whose

Table 1: **Distillation performance of EPIC and baselines in the inductive setting on three large-scale datasets.** The "#Layers" refers to the number of GNN-layers. The "Perf." refers to the performance, whose metric is Accuracy for `ogbn-arxiv` and `ogbn-products`, and ROC-AUC for `ogbn-proteins`. The "#Layers↓" refers to the relative decrease of numbers of graph convolutional layers compared to the teacher.

| Datasets | Teacher | | LSP-GCN | | GLNN | NOSMOG | EPIC (Ours) | | |
|---|---|---|---|---|---|---|---|---|---|
| | #Layers | Perf. | #Layers | Perf. | Perf. | Perf. | #Layers | #Layers↓ | Perf. |
| Arxiv | 28 | $72.91_{\pm 0.00}$ | 4 | $67.12_{\pm 0.33}$ | $56.16_{\pm 0.36}$ | $62.46_{\pm 0.39}$ | 3 | 89.29% | $\mathbf{73.06}_{\pm \mathbf{0.13}}$ |
| Products | 112 | $77.86_{\pm 0.00}$ | 4 | $72.43_{\pm 0.28}$ | $60.11_{\pm 0.06}$ | - | 7 | 93.75% | $\mathbf{78.58}_{\pm \mathbf{0.27}}$ |
| Proteins | 1,001 | $\mathbf{85.91}_{\pm \mathbf{0.08}}$ | 4 | - | $74.40_{\pm 0.09}$ | $60.63_{\pm 1.67}$ | 60 | 94.01% | $85.85_{\pm 0.09}$ |

runtime is marginal and space complexity $\mathcal{O}(|\mathcal{V}|)$ is similar to the node features $\mathbf{X}$. Therefore, the acceleration of inference without performance degradation under the transductive setting is trivial.

To distill deep GNNs on large-scale graphs and enforce a fair comparison, we select the datasets, the teacher models, the baselines, and the hyperparameters as follows.

**Datasets.** We evaluate the proposed EPIC on three large-scale OGB (Hu et al., 2020) datasets, i.e., `ogbn-arxiv`, `ogbn-products`, and `ogbn-proteins`. These datasets have more than 100,000 nodes and 1,000,000 edges, and have become standard large-scale benchmarks in recent years. For more details, please refer to Appendix A.1.

**Teacher Models.** For `ogbn-products` and `ogbn-proteins`, we select the one with the highest ranking on the public leaderboard among GNNs with over 100 layers. For the relatively smaller `ogbn-arxiv`, we select the one with the highest ranking on the public leaderboard among GNNs with over 10 layers. Based on these criteria, we select RevGCN-Deep (with 28 layers) (Li et al., 2021), RevGNN-112 (with 112 layers) (Li et al., 2021), and RevGNN-Deep (with 1,001 layers) (Li et al., 2021) as our teacher models on `ogbn-arxiv`, `ogbn-products`, and `ogbn-proteins`, respectively. For more details, please refer to Appendix A.3.

**Baselines.** We compare EPIC with three representative KD4GNN frameworks, i.e., LSP (GNNs-to-GNNs) (Yang et al., 2020b), GLNN (GNNs-to-MLPs) (Zhang et al., 2021), and NOSMOG (GNNs-to-MLPs) (Tian et al., 2022). For more details, please refer to Appendix A.4 and A.5.

**Hyperparameters.** We follow the most of hyperparameters used to train our teachers (Li et al., 2021), except for the additional hyperparameters in EPIC such as the width of the student $d_S$, the weight coefficients $\lambda$ and $\mu$, the number $K$ of concatenated embeddings, and the ratio $\gamma$ for selecting the number of student layers. For a fair comparison, We use the grid search to find the best hyperparameters for EPIC and the three baselines (please see Appendix A.4 and A.6 for more details).

## 6.2 MAIN RESULTS

**Distillation Performance of EPIC.** In this section, we refer to the student(s) trained with EPIC as EPIC(s). Table 1 reports the distillation performance of EPIC and baselines in the inductive setting on three large-scale datasets. The numbers of the EPIC student layers are selected in the narrowed range based on EPIC bounds. We train each teacher for only once, as training deep GNNs takes a long time. For RevGNN-Deep on `ogbn-proteins`, we report the mean and standard deviation by running 50 mini-batch inferences with different random seeds. For the students, we train each one for five times with different seeds, run 50 mini-batch inferences with different seeds for each trained student (we run full-batch inference if the GPU memory can afford it), and report the mean and standard deviation.

As shown in Table 1, EPIC reduces the numbers of teacher layers by at least 89.29%, while achieving comparable or even superior performance. Specifically, on `ogbn-arxiv` and `ogbn-products`, EPICs outperform their teachers with relative decreases of numbers of layers by 89.29% and 93.75%, respectively. On `ogbn-proteins`, the performance of EPIC is slightly lower than its 1,001-layer teacher, while its relative decrease of the number of layers is as high as 94%.

To further demonstrate the effectiveness of EPIC, we compare it with LSP (Yang et al., 2020b), GLNN (Zhang et al., 2021), and NOSMOG (Tian et al., 2022). For LSP, we follow the original paper (Yang et al., 2020b) to use 4-layer GCNs as students. We do not report the result of LSP-GCN on `ogbn-proteins`, as the vanilla GCN is difficult to encode graphs with edge features. For GLNN and NOSMOG, the students are MLPs. We do not report the result of NOS-MOG on `ogbn-products`, as it takes more than 50 hours to generate positional encodings for test nodes, which is much longer than the overall inference time of the teacher. As shown in Table 1, the performance of baselines in distilling deep GNNs is unsatisfactory, although they have achieved success in distilling moderate-sized GNNs. We attribute the reason to the weak expressive power of GCNs and MLPs. When distilling deep GNNs, the embeddings of teachers are extremely hard for them to approximate, hence the knowledge distillation becomes ineffective. Besides, in the inductive setting, shallow students do not have access to the teachers' soft labels on test nodes to use as guidance, which further poses challenges to knowledge distillation.

Table 2: **Comparisons of inference time between EPICs and deep teachers.** The unit of time in the table is seconds. The "Spd.↑" refers to the relative improvements of inference speed brought by EPIC.

| Datasets | Arxiv | Products | Proteins |
|---|---|---|---|
| Teacher | $0.52_{\pm 0.01}$ | $14.17_{\pm 0.05}$ | $113.29_{\pm 0.75}$ |
| EPIC | $0.08_{\pm 0.00}$ | $1.06_{\pm 0.04}$ | $14.11_{\pm 0.42}$ |
| Spd.↑ | $6.50\times$ | $13.37\times$ | $8.03\times$ |

Table 3: **The absolute improvements of performance brought by $\mathcal{L}_{\mathrm{SL}}$ and $\mathcal{L}_{\mathrm{EP}}$.** The improvements are denoted $\Delta_{\mathrm{SL}}$ and $\Delta_{\mathrm{EPIC}}$, respectively.

| Datasets | Arxiv | Products | Proteins |
|---|---|---|---|
| Vanilla EPIC | 72.91 | 77.71 | 85.00 |
| $\mathcal{L}_{\mathrm{SL}}$ | 72.94 | 77.83 | 85.25 |
| $\Delta_{\mathrm{SL}}$ | 0.03 | 0.12 | 0.25 |
| $\mathcal{L}_{\mathrm{SL}} + \mathcal{L}_{\mathrm{EP}}$ | 73.06 | 78.58 | 85.85 |
| $\Delta_{\mathrm{EPIC}}$ | 0.12 | 0.75 | 0.60 |

**Acceleration Effect of EPIC.** Table 2 shows the inference time of EPICs and their teachers. We report the mean and standard deviation by running inference for five times with different random seeds. For the deep teachers on `ogbn-arxiv`, `ogbn-products`, and `ogbn-proteins`, EPIC achieves speedups of $6.50\times$, $14.06\times$ and $8.03\times$, respectively. This demonstrates the effectiveness of EPIC in accelerating the inference of deep GNNs.

## 6.3 ANALYSIS OF EPIC BOUND

In this section, we conduct numerical experiments to empirically analyze the EPIC bound. We plot the line charts of the EPIC bound $u_{\mathrm{EPIC}}(M; L)$ and the log-EPIC bound $\log(u_{\mathrm{EPIC}}(M; L))$ with respect to the number of student layers $M$ on `ogbn-arxiv`, `ogbn-products`, and `ogbn-proteins`, as shown in Figure 1. We observe that $u_{\mathrm{EPIC}}(M; L)$ decreases rapidly when $M$ is small, and then remains at a low level. Besides, we observe that $\log(u_{\mathrm{EPIC}}(M; L))$ decreases linearly, which implies that $u_{\mathrm{EPIC}}(M; L)$ converges exponentially to zero (note that $u_{\mathrm{EPIC}}(L, L) = 0$) as $M$ increases. This empirically guarantees that we can distill deep GNNs into shallow GNNs.

## 6.4 ABLATION STUDIES

In this section, we conduct ablation studies to analyze the improvements of performance brought by $\mathcal{L}_{\mathrm{SL}}$ and $\mathcal{L}_{\mathrm{EP}}$. As shown in Table 3, both $\mathcal{L}_{\mathrm{SL}}$ and $\mathcal{L}_{\mathrm{EP}}$ bring improvements of performance to the EPIC students. Specifically, compared to vanilla EPIC students, $\mathcal{L}_{\mathrm{SL}}$ improves the performance by 0.03, 0.12, and 0.25 on `ogbn-arxiv`, `ogbn-products`, and `ogbn-proteins`, respectively. On this basis, $\mathcal{L}_{\mathrm{EP}}$ brings improvements of performance by 0.12, 0.75, and 0.60 on the three datasets, respectively. These results illustrate the effectiveness of the feature distillation loss and its potential in combination with other distillation loss terms.

## 7 CONCLUSION

In this paper, we propose the first GNN KD framework that quantitatively analyzes the teacher-student expressive power gap, namely EPIC. We show that the minimum error of approximating the teacher's embeddings with the student's embeddings has an upper bound, which decreases rapidly w.r.t. the number of student layers. We empirically demonstrate that the upper bound converges exponentially to zero as the number of student layers increases. Moreover, we propose to narrow the range of tuning the number of student layers based on the upper bound, and use an expressive power gap-related loss to further encourage the student to generate embeddings similar to those of the teacher. Experiments on large-scale benchmarks demonstrate that EPIC can effectively reduce the numbers of layers of deep GNNs, while achieving comparable or superior performance.

