# OpenReview forum: "EPIC: Compressing Deep GNNs via Expressive Power Gap-Induced Knowledge Distillation"
_ICLR.cc/2024/Conference — Submitted to ICLR 2024_

### Official Review · Reviewer_8ZML · 2023-10-17

**Soundness:** 3 good
**Presentation:** 3 good
**Contribution:** 2 fair
**Rating:** 5
**Confidence:** 4

**Summary:**

This paper theoretically and experimentally analyzes the expressive power gap between deep GNN teachers and lightweight students. It formulates the estimation of the expressive power gap to an embedding regression problem. The results on three large-scale datasets demonstrate the effectiveness of the proposed method. Overall, the paper is well written with an interesting topic.

**Strengths:**

- The analysis of graph knowledge distillation from the perspective of expressive power is novel.
- The paper is well written and presented, especially about the notation and background.
- EPIC outperforms 1001-layer GNNs with a 4-layer GNN.

**Weaknesses:**

- My main concern is the motivation for the paper. Although deep GNNs have been a popular research topic, deep GNNs are prone to suffer from overfitting and over-smoothing and so far have not been widely used as shallow GNNs. I am confused about the significance of studying distillation for deep GNN.
- The authors did not conduct their experiments in a transductive setting as GLNN and NOSMOG did.
- Why does the performance of NOSMOG drop so much in Table 1? According to their original paper, NOSMOG can achieve performance even better than the teacher GNN.
- Can the authors provide more results on the different layers of teacher GNN in Table 1? In particular, I would like to know how generalizable EPIC is in a shallow teacher GNN setting?
- As a GNN-to-GNN distillation method, the authors mainly compare EPIC with two GNN-to-MLP methods (GLNN and NOSMOG) in their experiments. Comparisons with some SOTA GNN-to-GNN distillation methods are missing, such as those introduced by the authors in related work.

**Questions:**

Please explain (3-5) in the weakness part.
How about the time complexity (especially the bound estimation) and running time?

**Details Of Ethics Concerns:**

No ethics review is needed.

---

> ### Author Response · Authors · 2023-11-21
> **Response to Reviewer 8ZML (1/2)**
>
> We thank the reviewer for the insightful and valuable comments. We respond to your comments as follows and sincerely hope that our rebuttal could properly address your concerns. If so, we would deeply appreciate it if you could raise your score. If not, please let us know your further concerns, and we will continue actively responding to your comments and improving our submission.
>
> **1. The motivation for studying the distillation for deep GNNs.**
>
> Thanks for your comments.
>
> We summarize the motivation for studying the distillation for deep GNNs as follows.
>
> - First, on large-scale datasets (e.g., ogbn-arxiv, ogbn-products, and ogbn-proteins), the deep teacher models that we distill in our paper (i.e., RevGCN-Deep, RevGNN-112, and RevGNN-Deep) outperform what existing KD4GNN works aim to distill, such as GCN, GraphSAGE, GAT, and APPNP. Compared with shallow GNNs, deep GNNs with more message passing layers are better at capturing long-range patterns [1, 2], which are critical for many downstream tasks such as search engine [3], recommendation systems [4], and molecular property prediction [5].
>
> - Second, although some deep GNNs still face challenges such as over-fitting and over-squashing, there have been many efforts to solve these challenges, such as [6, 7, 8, 9, 10, 11]. We believe that the challenges are temporary obstacles to the development of deep GNNs, and we believe that deep GNNs will receive widespread attention in the future.
> - Last but not least, the slow inference speed is also one of the important factors limiting the real applications of deep GNNs. Despite the great success in compressing and accelerating moderate-sized or shallow GNNs of existing KD techniques, it still remains a tough challenge to distill deep GNNs due to the huge expressive power gap between the teachers and students. Therefore, this paper aims to address the challenge of distilling deep GNNs that excel on large-scale graphs and possess numerous layers.
>
> [1] Chen, Tianlong, et al. "Bag of tricks for training deeper graph neural networks: A comprehensive benchmark study." *IEEE Transactions on Pattern Analysis and Machine Intelligence* 45.3 (2022): 2769-2781.
>
> [2] Cong, Weilin, Morteza Ramezani, and Mehrdad Mahdavi. "On provable benefits of depth in training graph convolutional networks." *Advances in Neural Information Processing Systems* 34 (2021): 9936-9949.
>
> [3] Brin, Sergey, and Lawrence Page. "The anatomy of a large-scale hypertextual web search engine." *Computer networks and ISDN systems* 30.1-7 (1998): 107-117.
>
> [4] Fan, Wenqi, et al. "Graph neural networks for social recommendation." *The world wide web conference*. 2019.
>
> [5] Kearnes, Steven, et al. "Molecular graph convolutions: moving beyond fingerprints." *Journal of computer-aided molecular design* 30 (2016): 595-608.
>
> [6] Li, Guohao, et al. "Training graph neural networks with 1000 layers." *International conference on machine learning*. PMLR, 2021.
>
> [7] Chen, Ming, et al. "Simple and deep graph convolutional networks." *International conference on machine learning*. PMLR, 2020.
>
> [8] Zhao, Lingxiao, and Leman Akoglu. "Pairnorm: Tackling oversmoothing in gnns." *arXiv preprint arXiv:1909.12223* (2019).
>
> [9] Guo, Xiaojun, et al. "Contranorm: A contrastive learning perspective on oversmoothing and beyond." *arXiv preprint arXiv:2303.06562* (2023).
>
> [10] Jaiswal, Ajay, et al. "Old can be gold: Better gradient flow can make vanilla-gcns great again." *Advances in Neural Information Processing Systems* 35 (2022): 7561-7574.
>
> **2. The experiments in the transductive setting.**
>
> Thanks for your comments. We conduct experiments in the setting of transductive training. The results are as follows.
>
> | Datasets | Teacher | LSP   | GLNN  | NOSMOG | EPIC (Ours) |
> | -------- | ------- | ----- | ----- | ------ | ----------- |
> | Arxiv    | 72.96   | 69.77 | 57.41 | 65.79  | 73.33       |
> | Products | 82.11   | 78.31 | 61.45 | 64.62  | 83.73       |
> | Proteins | 87.42   | -     | 75.19 | 73.84  | 87.13       |
>
> **3. The drop of the performance of NOSMOG compared to that reported in the original paper.**
>
> Thanks for your comments.
>
> We train MLPs using the source code of NOSMOG in the inductive setting. We speculate that the reasons for the drop of performance of NOSMOG compared to that reported in the original paper are as follows.
>
> In the original paper, the authors train NOSMOG in the transductive setting and a loose inductive setting (with only 20% of test nodes invisible to the models). In these settings, the teacher model's predictions on most test nodes are available to students, hence the students can directly fit the predictions of their teachers on test nodes to achieve comparable or even superior performance to the teachers. However, we conduct experiments in the strictly inductive setting, which is more challenging for student MLPs than the experiment settings in the original paper.

---

> > ### Author Response · Authors · 2023-11-21
> > **Response to Reviewer 8ZML (2/2)**
> >
> > **4. The distillation results for shallow GNN teachers.**
> >
> > Thanks for your comments.
> >
> > We conduct experiments of distilling shallow GNN teachers on ogbn-arxiv in the transductive and inductive settings. Specifically, in the two settings, we train teacher models with 7 and 3 layers, and distill them to 3-layer and 1-layer students, and a 1-layer student, respectively. The results are as follows.
> >
> > | Setting            | Teacher-7layers | EPIC-3layers | EPIC-1layer | Teacher-3layers | EPIC-1layer |
> > | ------------------ | --------------- | ------------ | ----------- | --------------- | ----------- |
> > | ***Transductive*** | 70.93           | 72.31        | 70.59       | 70.35           | 70.57       |
> > | ***Inductive***    | 70.69           | 71.91        | 70.42       | 69.97           | 70.13       |
> >
> > **5. Comparisons between EPIC and the state-of-the-art GNNs-to-GNNs distillation method.**
> >
> > Thanks for your comments.
> >
> > Because different GNNs-to-GNNs distillation works use different teacher models and conduct experiments on different datasets, there is currently no widely recognized state-of-the-art GNNs-to-GNNs distillation framework. Therefore, we select a new GNNs-to-GNNs distillation baseline, i.e., BGNN [1] with 2-layer GCNs as students, as the average results of BGNN on the datasets Cora, CiteSeer, and Pubmed are the best among existing GNNs-to-GNNs frameworks to the best of our knowledge. Besides, we implement BGNN ourselves, as BGNN is not open source. If you want us to compare EPIC with other baselines, please let us know and we will add new baselines to our paper. The results are as follows.
> >
> > | Datasets         | Teacher | BGNN-GCN | EPIC (Ours) |
> > | ---------------- | ------- | -------- | ----------- |
> > | **Inductive**    |         |          |             |
> > | Arxiv            | 72.91   | 68.87    | 73.06       |
> > | Products         | 77.86   | 74.01    | 78.58       |
> > | Proteins         | 85.91   | -        | 85.85       |
> > | **Transductive** |         |          |             |
> > | Arxiv            | 72.96   | 69.94    | 73.33       |
> > | Products         | 82.11   | 79.45    | 83.73       |
> > | Proteins         | 87.42   | -        | 87.13       |
> >
> > [1] Guo, Zhichun, et al. "Boosting graph neural networks via adaptive knowledge distillation." *Proceedings of the AAAI Conference on Artificial Intelligence*. Vol. 37. No. 6. 2023.

---

> ### Author Response · Authors · 2023-11-23
> **We are looking forward to your further comments.**
>
> Dear Reviewer 8ZML,
>
> Thank you again for your careful reading and insightful comments, which are of great significance for improving our work. The deadline for the discussion stage is approaching, and we are looking forward to your feedback and/or questions. We sincerely hope that our rebuttal has properly addressed your concerns. If so, we would deeply appreciate it if you could raise your score. If not, please let us know your further concerns, and we will continue actively responding to your comments and improving our submission.
>
> Best,
>
> Authors

---

### Official Review · Reviewer_ZSeG · 2023-10-30

**Soundness:** 3 good
**Presentation:** 2 fair
**Contribution:** 2 fair
**Rating:** 5
**Confidence:** 4

**Summary:**

This paper focuses on the knowledge distillation of deep GNNs, such as that with over 100 layers. To address the issue of the representation ability gap between teacher and student networks, this paper proposes a framework, EPIC, leveraging the embedding regression based on the theory of polynomial approximation. Specifically, EPIC first shows the EPIC bound exponentially converges as the number of student layers increases with experiments. Then, with this observation, it selects an appropriate layer number of the student network by analyzing the EPIC bound.  Furthermore, an expressive power gap-induced loss term is proposed to reduce the gap. In the experiments, it reduces 94% of layers, improves 8x speed, and obtains comparable performance for 1,001-layer RevGNN-Deep.

**Strengths:**

1. The field of distilling a very deep GNN model is less explored. By distilling, the inference speed can be improved by a large margin, which benefits the potential applications.
2. A theoretical framework is proposed to analyze the gap between the student and teacher network from a spectral perspective.
3. The theoretical results with an EPIC bound are used to select the appropriate layer number. It is interesting and meaningful to reduce the cost of the hyper-parameter tuning.

**Weaknesses:**

The main concerns lie in the method details and experimental settings. The details are shown below:
1. The hyper-parameter $\gamma$ in Eq. (8) weakens (somehow) the theoretical results. The optimal # layers $M^*$ are dependent on this hyper-parameter, regardless of the EPIC bound. How to choose $\gamma$, is there any guidance?   Does it involve the extra hyperparameter tuning cost?  When comparing tuning M (layer number) and $\gamma$, what is the advantage of the latter?
2. The scalability is also a concern, since it requires eigenvalue decomposition to find out all the eigenvalues, incurring high memory and time complexity, especially for large-scale graphs, which is the setting in this work. It is suggested to add some experimental cost to the process of calculating EPIC bounds.
3. The singular value decomposition (SVD) and Laplacian requires the assumptions of linearity.  I understand that this may be easier for derivation. However, it still would be helpful if you could explain why this linear-based theory could be used for non-linear graph models, such as GCN and GAT.  It seems that the used teacher model is also non-linear.
4. Novelty concerns on the EPIC loss.  The paper claims the new EPIC loss (expressive power gap-induced loss). However, If I understand correctly, H_S and H_T are representations of student and teacher models respectively. It seems the same as standard knowledge-distilling loss where the embeddings in the student network mimic the ones in the teacher network. It is suggested to mention this and avoid the new terms if they are similar. Otherwise, it is better to clarify the differences and provide corresponding evidence (insights and experimental results.)
5. There are some concerns in the experiment design. First, the comparison needs to be more fair. As for GLNN and NOSMOG, the student model is MLP, while for EPIC, the student model is a GNN, the performance gain may be caused by the enhanced expressivity of student models. Second, this experiment can't be used to verify the correctness of the proposed theorem (i.e., is this bound tight when deciding the # of layers in student GNNs). Do the results of the real experiments align with the curves in Figure 1?  I suggest redesigning this experiment to stress the correctness of the proposed theorem. Third, It is suggested to add baselines that train the GNN with fewer layers from scratch.  Again, the cost of hyper-parameter $\gamma$ is an issue.
6. (Minor), It is suggested to refine Figures 1 and 2.  For example, it is better to provide more details in Figure 2. Currently, the details of EPIC and Bound and EPIC loss are lost.

**Questions:**

1) Regard the $\gamma$. How to choose $\gamma$ in the experiments? Does it involve the extra hyperparameter tuning cost (How much)?  When comparing tuning M (layer number) and $\gamma$, what is the advantage of the latter?

2) What is the experimental cost of the calculating process of EPIC bounds?

3) Could you please explain why the proposed linear-based theory could be used for non-linear graph models?

4) What is the unique contribution of the proposed EPIC loss?

---

> ### Author Response · Authors · 2023-11-21
> **Response to Reviewer ZSeG**
>
> We thank the reviewer for the insightful and valuable comments. We respond to your comments as follows and sincerely hope that our rebuttal could properly address your concerns. If so, we would deeply appreciate it if you could raise your score. If not, please let us know your further concerns, and we will continue actively responding to your comments and improving our submission.
>
> **1. The selection of the hyperparameter $\gamma$ and the corresponding cost.**
>
> Thanks for your comments.
>
> After reading your comments, we remove the hyperparamter $\gamma$ and instead use the EPIC bounds to narrow the range of tuning the number of student layers as follows.
>
> 1. We compute the EPIC bounds $(u_{\rm EPIC}(M;L))_{M=1}^{L-1}$ .
> 2. We plot $u_{\rm EPIC}(M;L)$ as a function of $M$, as shown in Figure 1.
> 3. By observing the plot, we find the maximum value $M_{\max}$ such that $u_{\rm EPIC}(M;L)$ decreases slowly when $M>M_{\max}$.
> 4. We tune the hyperparamter $M$ in $[1, M_{\max}]$.
>
> Because $u_{\rm EPIC}(M;L)$ converges exponentially to zero as $M$ increases, $M_{\rm max} $ is usually much smaller than $L$. Thus, we can significantly reduce the range of tuning $M$.
>
> **2. The complexity of computing the EPIC bound is high.**
>
> Thanks for your comments.
>
> Please note that we do not need to compute EPIC bounds in the inference stage of GNNs, hence the computation of EPIC bounds do not affect the inference speed of GNNs, which is the focus of this paper and most knowledge distillation methods.
>
> **3. The relationship of our linear-based theory to non-linear graph models.**
>
> Thanks for your comments.
>
> The reason why we analyze the expressive power of GNNs from a linear perspective is that this is a standard practice [1, 2], and that existing works [1, 3] have demonstrated that the theoretical analysis under the linear assumption is applicable to general non-linear GNNs.
>
> [1] Xu, Keyulu, et al. "Optimization of graph neural networks: Implicit acceleration by skip connections and more depth." *International Conference on Machine Learning*. PMLR, 2021.
>
> [2] Wang, Xiyuan, and Muhan Zhang. "How powerful are spectral graph neural networks." *International Conference on Machine Learning*. PMLR, 2022.
>
> [3] Thekumparampil, Kiran K., et al. "Attention-based graph neural network for semi-supervised learning." *arXiv preprint arXiv:1803.03735* (2018).
>
> **4. The comparison between EPIC and existing GNNs-to-MLPs distillation frameworks is unfair.**
>
> Thanks for your comments.
>
> Because we focus on how to select an appropriate student that is lightweight, while expressive, and MLP is currently a popular student structure for distilling GNNs, we compare our proposed EPIC with state-of-the-art GNNs-to-MLPs distillation frameworks. As mentioned in Section 6.2, we attribute the reason for the unsatisfactory performance of GNNs-to-MLPs frameworks to the weak expressive power of MLPs. The experiments support our claim in Introduction that "MLPs are not 'good students' for distilling deep GNNs".
>
> **5. Experiments of baselines that train the GNN with fewer layers from scratch.**
>
> Thanks for your comments.
>
> We conduct experiments of training student GNNs with fewer layers from scratch on the three OGB datasets. The results are as follows.
>
> | Arxiv  | 2     | 3     | 4     | 5     | 6     | 7     |
> | :------ | :-----: | ----- | ----- | ----- | ----- | ----- |
> | w/ KD  | 72.57 | 73.06 | 73.18 | 73.24 | 73.35 | 73.24 |
> | w/o KD | 72.25 | 72.91 | 73.00 | 72.65 | 72.34 | 72.15 |
>
> | Products | 2     | 3     | 4     | 5     | 6     | 7     |
> | :-------- | ----- | ----- | ----- | ----- | ----- | ----- |
> | w/ KD    | 76.80 | 76.99 | 77.70 | 78.32 | 78.26 | 78.58 |
> | w/o KD   | 76.67 | 76.86 | 76.85 | 77.29 | 77.56 | 77.71 |
>
> | Proteins | 10    | 20    | 30    | 40    | 50    | 60    |
> | -------- | ----- | ----- | ----- | ----- | ----- | ----- |
> | w/ KD    | 85.64 | 85.68 | 85.33 | 85.81 | 85.72 | 85.85 |
> | w/o KD   | 84.13 | 84.23 | 84.58 | 84.75 | 84.89 | 85.00 |

---

> > ### Comment · Reviewer_ZSeG · 2023-11-22
> > **Thanks for the response.**
> >
> > Thank you for your response, and I apologize for my delayed reply.
> >
> > Upon review, I find that the current revision does not adequately address my concerns. Specifically:
> >
> > 1) Clarification on 'h' Removal: The implications of removing 'h' are not clear. Could you elaborate on how this affects the study's results?
> >
> > 2) Rationale Behind Unfair Comparison: The justification for using an unfair comparison remains unconvincing. A more thorough explanation is needed to understand this choice.
> >
> > 3) Overlooked Novelty Aspects: The issue of novelty in your study appears to have been overlooked. It's important to distinctly highlight what sets your research apart.
> >
> > 4) Discrepancies in Revision: I did not find any changes in the revised paper that correspond to the issues raised in the response.
> >
> > Given these unresolved issues, my evaluation of the paper remains unchanged.

---

> > > ### Author Response · Authors · 2023-11-23
> > > **Thank you for your further comments.**
> > >
> > > We thank the reviewer again for the constructive comments. We respond to your comments as follows and sincerely hope that our rebuttal could properly address your concerns. If so, we would deeply appreciate it if you could raise your score. If not, please let us know your further concerns, and we will continue actively responding to your comments and improving our submission.
> > >
> > > **1. The implications of removing $\gamma$ is unclear. How does this affect the results?**
> > >
> > > Removing the hyperparameter $\gamma$ and directly tuning $M$ does not affect the results of distillation, but is more efficient. This is because $\gamma$ is continuous, while $M$ is discrete.
> > >
> > > Specifically, in the original version of our paper, the mapping $\gamma \mapsto M^*$ is not injective, that is, there are many different $\gamma$ corresponding to the same $M^*$. Hence, we need to first find an appropriate $\gamma$ in the continuous $(0, 1]$, and compute $M^*$, which results in the high cost of tuning hyperparameters.
> > >
> > > After removing $\gamma$, we only need to tune $M$ in the discrete set {$1,2,\ldots, M_{\max}$}, which is more efficient than tuning $\gamma$.
> > >
> > > **2. The explanation about the comparisons between EPIC and existing GNNs-to-MLPs frameworks.**
> > >
> > > The explanation is as follows.
> > >
> > > First, the selection of student models is a critical factor for knowledge distillation [1, 2]. Existing GNNs-to-MLPs works imply the view that "MLPs are enough for the distillation of GNNs" and have triggered an upsurge in the GNNs-to-MLPs distillation. However, we claim that this view does not apply to the distillation for deep GNNs, and call for more attention to turn to the research on distilling deep GNNs using GNNs as students. Therefore, we select two GNNs-to-MLPs frameworks as baselines to support our claim.
> > >
> > > Second, we fully understand that the reason why the reviewer thinks the comparison is unfair, i.e., our student models are more expressive than MLPs, leading to the difference in the performance of distillation. However, this is exactly the claim we want to support through our experiments.
> > >
> > > [1] Mirzadeh, Seyed Iman, et al. "Improved knowledge distillation via teacher assistant." *Proceedings of the AAAI conference on artificial intelligence*. Vol. 34. No. 04. 2020.
> > >
> > > [2] Gao, Mengya, Yujun Wang, and Liang Wan. "Residual error based knowledge distillation." *Neurocomputing* 433 (2021): 154-161.
> > >
> > > **3. The contributions of our paper.**
> > >
> > > We summarize our contributions as follows.
> > >
> > > 1. **The problem we focus on (i.e., distillation for deep GNNs) is more challenging than what existing works focus on (i.e., distillation for moderate-sized GNNs).** Specifically, this is the ***first*** paper that attempts to distill GNNs with more than 100 layers to the best of our knowledge. It is challenging to distill deep GNNs due to the difficult trade-off between the needs of being "lightweight" and being "expressive" when selecting a student for the deep teacher.
> > > 2. **Our theoretical analysis and experiments reveal that it is feasible to distill deep GNNs into shallow GNNs.**
> > >    - **This is the *first* paper that quantitatively analyzes the teacher-student expressive power gap from a theoretical perspective.** Our theoretical analysis reveals the feasibility of distilling a deep GNN into a shallow GNN, as there exist some shallow GNNs that have similar expressive power to the deep GNN. This conclusion is significant for the distillation of deep GNNs and is surprising in a way, as the size of the neighborhood involved in the computation of each node is very different between deep GNNs and shallow GNNs.
> > >    - **We reduce the number of layers of the 1,001-layer RevGNN-Deep by 94% and accelerate its inference by roughly eight times, while achieving comparable performance.** The result shows that it is possible to significantly reduce the number of layers of deep GNNs via knowledge distillation, bringing confidence to the future research on knowledge distillation for deep GNNs.
> > >
> > > **4. The changes in the revised version of our submission.**
> > >
> > > Thanks for the reminder. We have revised our paper based on your comments. The changes are marked in red. For the revision in appendices, please refer to the supplementary materials. For your convenience, we list the changes as follows.
> > >
> > > 1. In Abstract, Introduction, Section 5.1, and Conclusion, we revise the contents about the method to "narrow the range of tuning the number of student layers".
> > > 2. In Page 6, Section 5.1, we revise the content about the computation of EPIC bounds.
> > > 3. In Appendix B.1, we add the relationship of linear-based theory to non-linear GNNs.
> > > 4. In Appendix A.5, we add the explanation about why do we compare EPIC with GNNs-to-MLPs frameworks.
> > > 5. In Appendix C.1, we add the experiments of training GNNs with fewer layers from scratch.
> > > 6. We revise all statements about EPIC loss to "a feature distillation loss" or "an expressive power gap-related loss".

---

### Official Review · Reviewer_G7rf · 2023-10-30

**Soundness:** 2 fair
**Presentation:** 2 fair
**Contribution:** 2 fair
**Rating:** 3
**Confidence:** 4

**Summary:**

The author(s) theoretically demonstrate that the shallow student model lacks sufficient expressive power to mimic the teacher;
hence, distilling deep GNNs remains a tough challenge.
The derived upper bound allows for quantitative analysis of the gap, making it easy to determine an appropriate number of layers for student models.

**Strengths:**

1. Distilling very deep GNNs is challenging and important.
2. This work provides some valuable suggestions on layer number selection.

**Weaknesses:**

1. The conclusion obtained from Theorem 1 is obvious, and the upper bound derived therein appears difficult to apply in practice.
While the hyperparameter $\gamma$ can achieve this to some extent, the value of $\gamma$ is not directly related to classification performance gap. Therefore, a search is also required to determine the appropriate value.
2. The complexity of computing EPIC bounds should be analyzed (since SVD for large-scale graphs is expensive).
3. Feature distillation (namely the EPIC loss presented in this paper) is a well-known technique.

**Questions:**

Please refer to Weaknesses.

---

> ### Author Response · Authors · 2023-11-21
> **Response to Reviewer G7rf**
>
> We thank the reviewer for the insightful and valuable comments. We respond to your comments as follows and sincerely hope that our rebuttal could properly address your concerns. If so, we would deeply appreciate it if you could raise your score. If not, please let us know your further concerns, and we will continue actively responding to your comments and improving our submission.
>
> **1. The conclusion obtained from Theorem 1 is obvious.**
>
> Thanks for your comments.
>
> Theorem 1 guarantees that we can distill a deep GNN into a shallow GNN, as there exist some shallow GNNs that have similar expressive power to the deep GNN. This conclusion is significant for the distillation of deep GNNs and is surprising in a way, as the numbers of neighbor hops included in the computation of each node in deep GNNs and shallow GNNs are very different.
>
> Specifically, Theorem 1 shows that the minimum approximation error of $\mathbf{H}_S^{(M)}$ to $\mathbf{H}_T^{(L)}$ in terms of the Frobenius norm has an upper bound that decreases monotonically with respect to $M$. Furthermore, we empirically demonstrate that the upper bound converges exponentially to zero as $M$ increases.
>
> **2. How to apply the EPIC bound and select the hyperparameter $\gamma$ in practice?**
>
> Thanks for your comments.
>
> After reading your comments, we remove the hyperparamter $\gamma$ and instead use the EPIC bounds to narrow the range of tuning the number of student layers as follows.
>
> 1. We compute the EPIC bounds $(u_{\rm EPIC}(M;L))_{M=1}^{L-1}$ .
> 2. We plot $u_{\rm EPIC}(M;L)$ as a function of $M$, as shown in Figure 1.
> 3. By observing the plot, we find the maximum value $M_{\max}$ such that $u_{\rm EPIC}(M;L)$ decreases slowly when $M>M_{\max}$.
> 4. We tune the hyperparamter $M$ in $[1, M_{\max}]$.
>
> Because $u_{\rm EPIC}(M;L)$ converges exponentially to zero as $M$ increases, $M_{\rm max} $ is usually much smaller than $L$. Thus, we can significantly reduce the range of tuning $M$.
>
> **3. The complexity of computing the EPIC bound is high.**
>
> Thanks for your comments.
>
> Please note that we do not need to compute EPIC bounds in the inference stage of GNNs, hence the computation of EPIC bounds does not affect the inference speed of GNNs, which is the focus of this paper and most knowledge distillation methods.

---

> ### Author Response · Authors · 2023-11-23
> **We are looking forward to your further comments.**
>
> Dear Reviewer G7rf,
>
> Thank you again for your careful reading and insightful comments, which are of great significance for improving our work. The deadline for the discussion stage is approaching, and we are looking forward to your feedback and/or questions. We sincerely hope that our rebuttal has properly addressed your concerns. If so, we would deeply appreciate it if you could raise your score. If not, please let us know your further concerns, and we will continue actively responding to your comments and improving our submission.
>
> Best,
>
> Authors

---

### Meta-Review · Area_Chair_1dCU · 2023-12-15

**Metareview:**

This paper proposes EPIC, an expressive power-based method to distill knowledge from a deeper graph neural network (GNN) teacher model into a shallower GNN student model. The reviewers raised several critical concerns regarding the novelty, theoretical analysis, experiment design, and evaluation of the proposed method:

First, the key concept behind EPIC seems similar to common knowledge distillation methods that minimize the difference between student and teacher representations. The claim of a "new loss" needs more clarification on how it differs from existing techniques.

Second, the theoretical analysis relies on assumptions of linearity while common GNN models are highly non-linear. The bound may not accurately reflect real model capacities. Also, the bound computation requires expensive SVD, hurting scalability.

Additionally, the experiments compare EPIC against student MLPs, while it uses a GNN student itself. More comparisons with state-of-the-art GNN-to-GNN distillation methods are needed to showcase advantage. Experiments also do not strongly verify the theory.

Finally, motivation behind studying deep GNN distillation is unclear, as shallow GNNs remain dominant in practice. The transductive experimental setting used by existing methods is not followed.

I would encourage the authors to consider a major revision, and the paper could be much stronger if these concerns could be properly addressed.

**Justification For Why Not Higher Score:**

NA

**Justification For Why Not Lower Score:**

NA

---

### Decision · Program_Chairs · 2024-01-16

Reject